# Variants of HCMV *UL18* Sequenced Directly from Clinical Specimens Associate with Antibody and T-Cell Responses to HCMV

**DOI:** 10.3390/ijms232112911

**Published:** 2022-10-26

**Authors:** Shelley Waters, Silvia Lee, Ibnu Ariyanto, Shay Leary, Kylie Munyard, Silvana Gaudieri, Ashley Irish, Richard J. N. Allcock, Patricia Price

**Affiliations:** 1Curtin Health Innovation Research Institute, Curtin Medical School, Curtin University, Bentley 6102, Australia; 2PathWest Laboratory Medicine WA, Department of Microbiology, Nedlands 6009, Australia; 3Virology and Cancer Pathobiology Research Center, Faculty of Medicine, Universitas Indonesia, Jakarta 10430, Indonesia; 4Institute for Immunology and Infectious Diseases, Murdoch University, Murdoch 6150, Australia; 5School of Human Sciences, University of Western Australia, Nedlands 6009, Australia; 6Department of Medicine, Division of Infectious Diseases, Vanderbilt University Medical Center, Nashville, TN 37232, USA; 7Department of Nephrology, Fiona Stanley Hospital, Murdoch 6150, Australia; 8School of Biomedical Sciences, University of Western Australia, Nedlands 6009, Australia; 9PathWest Laboratory Medicine WA, Department of Diagnostic Genomics, Nedlands 6009, Australia

**Keywords:** cytomegalovirus, *UL18*, renal transplant recipients, people with HIV, deep sequencing

## Abstract

Around 80% of adults worldwide carry human cytomegaloviris (HCMV). The HCMV gene *UL18* is a homolog of HLA class I genes and encodes a protein with high affinity for the NK and T-cell cytotoxicity inhibitor LIR-1. *UL18* was deep sequenced from blood, saliva or urine from Indonesian people with HIV (PWH) (*n* = 28), Australian renal transplant recipients (RTR) (*n* = 21), healthy adults (*n* = 7) and neonates (*n* = 4). 95% of samples contained more than one variant of HCMV *UL18*, as defined by carriage of nonsynonymous variations. When aligned with immunological markers of the host’s burden of HCMV, the S318N variation associated with high levels of antibody reactive with HCMV lysate in PWH over 12 months on antiretroviral therapy. The A107T variation associated with HCMV antibody levels and inflammatory biomarkers in PWH at early timepoints. Variants D32G, D248N, V250A and E252D aligned with elevated HCMV antibody levels in RTR, while M191K, E196Q and F165L were associated with HCMV-reactive T-cells and proportions of Vδ2^−^ γδ T-cells—populations linked with high burdens of HCMV. We conclude that *UL18* is a highly variable gene, where variation may alter the persistent burden of HCMV and/or the host response to that burden.

## 1. Introduction

Human cytomegalovirus (HCMV) is a beta-herpesvirus carried by ~80% of all adults [1]. Primary infections are usually asymptomatic, but elevated levels of HCMV-reactive antibodies have been associated with accelerated development of cardiovascular disease (CVD) [2]. HCMV frequently causes congenital infections and is the leading non-genetic cause of sensorineural hearing loss in children [3]. It can also cause morbidity, neurodevelopment delays and vision impairment due to damage of the central nervous system [4]. Genetic diversity has been demonstrated in HCMV amplified from neonates [5]. HCMV infections occur in 20–60% of solid organ transplant recipients. In renal transplant recipients (RTR), these are associated with graft rejection, secondary bacterial or fungal infections and CVD, with risk affected by the source of the kidney, immunosuppressive therapies and prophylactic regimes [6]. HCMV-seronegative RTR are at a high risk of complications from primary HCMV infection which can result in graft rejection, organ invasive disease and mortality [7].

Most people with HIV (PWH) are HCMV seropositive [8,9,10]. HCMV in PWH can cause retinitis, as an AIDS-defining illness [11], but this is rare with antiretroviral therapy (ART). However, PWH maintains higher levels of HCMV-reactive antibodies than are seen in the general population [12]. Higher antibody levels are associated with CVD and cerebrovascular diseases in PWH [13].

HCMV has a large genome of 235 kb [14] which encodes 165–252 open reading frames (ORFs) depending on the strain. Only 45 ORFs are essential for HCMV replication in vitro, while others are involved in immunomodulation including homologs of host genes acquired through co-evolution with the host [15,16,17,18]. A well-documented example is *UL18* which is homologous to human leukocyte antigen (HLA) [19]. While *UL18* is not essential for HCMV replication in vitro, it has been retained in all clinical isolates that have been analysed to date (e.g., [20,21])

UL18 has ~21% amino acid sequence identity to the HLA class I molecules but its similar secondary and tertiary structure enables interactions with host proteins β_2_-microglobulin (B_2_M) and leukocyte immunoglobulin-like receptor-1 (LIR-1 aka LILRB1, ILT2, CD85) [22]. UL18 is only functional when bound to β2-microglobulin (B_2_M). LIR-1 is expressed on monocytes, dendritic cell, B-cells, T-cells and natural killer (NK) cells and functions as an inhibitory receptor [23]. UL18 binds to LIR-1 > 1000-fold higher affinity than the HLA-A2 receptor pair [24]. LIR-1 signalling inhibits cellular cytotoxicity, cytokine production and antigen-presenting cell activation. Furthermore, LIR-1 expression on NK cells is increased in the presence of HCMV DNA and antibodies in samples from RTR [25]. Deletion of the *UL18* gene from the laboratory-adapted HCMV strain AD169 decreased interferon (IFN)-α and IFN-γ production in co-cultures of blood leukocytes from healthy donors and infected fibroblasts [26].

Many publications examining HCMV diversity are based on Sanger sequencing of PCR amplicons, which may miss mixed infections [27,28,29]. Some studies have used HCMV propagated in vitro which may miss strains present only in vivo. Here, we describe an amplicon-enriched PCR protocol using high-resolution deep sequencing technologies to sequence directly from clinical samples. We compare *UL18* sequences from RTR, PWH, healthy adults and neonates with a Toledo reference strain derived from the urine of a congenitally infected child [30]. The RTRs were clinically stable and recruited in Western Australia more than two years after transplantation. Biomarkers of their HCMV burden and cardiovascular health have been described [31]. PWH were studied from the JakCCANDO project—a longitudinal study of HIV^+^ individuals commencing ART and monitored for 12 months in Jakarta, Indonesia. Cellular and serological markers of inflammation and their exceptionally high burden of HCMV were described [32].

## 2. Results

### 2.1. UL18 Is Highly Variable in Clinical Samples

Sequences targeting HCMV *UL18* were sought from 60 clinical samples (blood, saliva, or urine) and had an average mean read depth of 11,734. Twenty-eight samples were from Indonesian PWH (21 buffy coat and 7 saliva) collected after 0 to 3 months on ART, 21 were from Australian RTR (>2 years after transplant; 8 buffy coat and 13 saliva), 7 were from healthy Australian adults (2 buffy coat and 5 saliva), and 4 were from Australian neonates (urine). No adult donors had symptomatic infections.

Compared with Toledo (GenBank no. GU937742.1), there were 304 sites of nucleotide variation (Figure 1A), of which 64 sites were nonsynonymous substitutions (Figure 1B). These nonsynonymous substitutions had depth of > 50 reads, comprised >10% of the sequences from any individual and occurred in at least three samples. Of the 60 samples sequenced, 57 (95%) contained more than one nonsynonymous variant of HCMV *UL18*. None of the remaining three samples encoded a strain identical to the Toledo reference.

Variants at amino acid positions 29, 32, 32, 54 and 107 are located within the α1 domain of UL18. Variants 29, 31 and 32 are at the end of the β1 sheet. Variant 54 falls within the β3 sheet and 107 is at the end of the α1 helices. Variants at amino acid positions 181, 185, 191 and 196 are located within the α2 domain helices. Variants at amino acid positions 231, 248, 250, 252 and 265 are located within the α3 domain. Variant 231 is located within the B region of the α3 domain. Positions 248, 250 and 252 are located between the C and D region of α3 domain and variant 265 is between the D and E region. No variants altered an N-glycosylation site (23). Variations at sites 32, 107, 181, 185, 191, 196, 248. 250, 252 and 265 have been described previously (23).

### 2.2. Several Variations Were Group- and Sample-Specific

*UL18* sequences from neonates (*n* = 4) had eleven nonsynonymous variations, while sequences from adults (*n* = 56) had 64 variations (Table 1), including 53 seen only in adults. *UL18* sequences from Australian adults had 63 nonsynonymous variations, including 34 restricted to Australian samples. HCMV sequences from Indonesians had 42 nonsynonymous variations, including three restricted to Indonesian samples.

*UL18* sequences from buffy coat samples had 62 nonsynonymous variations, including one that was not found in urine or saliva. Sequences from saliva had 63 nonsynonymous variations, with five unique to saliva. T4M, T8S, L12M, K60R, E103L, M191K, E196Q, E295G and S318N were present in all groups and all sample types. H220Q was the most common variant present in ~91% (53/58) of samples. S318N, M191K and E196Q were present in ~85% (51/60), 80% (45/57) and 80% (45/57) of samples, respectively.

### 2.3. UL18 Variations Associate with HCMV Antibody Levels in PWH Starting ART in Indonesia, but Not in Australian RTR

We next assessed associations between *UL18* variants and levels of HCMV-reactive antibody [determined using a lysate of infected cells and HCMV IE-1 protein] in Indonesian PWH starting ART in the JakCCANDO cohort [33]. We have documented significant rises in CD4^+^ T-cell counts and HCMV lysate antibody levels in visits over 12 months on ART (V0–V12).

At position 318, 6/21 samples encoded only S, 5/21 carried both S and N variants and 10/21 carried only N. Carriage of the N variant was associated with lower levels of HCMV lysate antibody in PWH throughout the first year on ART (Figure 2A). Plasma levels of soluble IFNαR and CRP were significantly lower before ART and after 1 month on treatment (Figure 2B,C) in PWH carrying the N variant.

HCMV encoding N at position 318 (S318N) was also present in Australian RTR, with 11/15 individuals carrying N and 4/15 individuals carrying both N and S at this position. All six Australian healthy adults carried only N at position 318. Amongst RTR, position 318 did not align with significant differences in HCMV antibody levels (data not shown, *p* > 0.5).

At position 107, 6/21 PWH carried only the A variant, 8/21 carried both A and T, 7/21 carried only the T variant and none carried the P variant. PWH carrying only T had slightly higher levels of HCMV lysate-reactive antibody after 0–3 months on ART. Significant differences are presented (Figure 3A–C). Accordingly, plasma levels of soluble IFNαR and CRP were elevated in individuals with the T variant (Figure 3D,E), marking immune activation and viral replication.

HCMV encoding T at position 107 was present in Australian RTR. Eleven of 15 individuals carried only A, whilst 4/15 individuals both A and T. Only 1/6 healthy adults carried the T variant, 3/6 carried A alone, while the other 3/6 also carried a variant encoding P. 1/15 RTR also carried the P variant. Neither the A or T variants align with HCMV antibody levels in RTR (data not shown, *p* > 0.8).

Several other *UL18* variants are associated with levels of HCMV antibodies and/or CD4^+^ T-cell counts in PWH at isolated time points. These are summarized in Table 2. Analyses were performed in way that ensured that no group had n < 3. Mixed samples indicated by L/M. Carriage of the M variant at position 12 (L12M) was associated with reduced levels of HCMV lysate antibodies after 12 months on ART. Accordingly, populations of CD4^+^ T-cells were also reduced in individuals carrying the M variant after 6 and 12 months.

At position 31, 17/21 individuals carried F and 4/21 carried both F and Y. Carriage of the Y variant was associated with lower HCMV lysate antibody levels [*p* = 0.012 at V1 (Table 2), *p*= 0.29–0.56 at other timepoints]. Antibody levels did not rise on ART in individuals carrying both F and Y.

At position 54, the Y variant was associated with reduced levels of HCMV IE-1 antibodies [*p* = 0.028 at V3 (Table 2), *p* = 0.11–0.69 at other timepoints]. The pattern suggests a delayed rise in antibody levels on ART.

The G181E variant was carried in linkage disequilibrium with I185T. Individuals with the G variant at position 181 had higher HCMV IE-1 antibody levels after 12 months than those carrying both G and E (Table 2, *p* = 0.038).

Individuals with HCMV encoding only M at position 191 (M191K) had higher levels of HCMV lysate antibody [*p* = 0.03 at V3 (Table 2), *p* = 0.14 (V1)–0.9 (V0) data not shown]. Higher levels were also seen in PWH carrying A at position 231 (S231A) [*p* = 0.05 at V12 (Table 2), *p* = 0.07 (V6)–0.96 (V0); data not shown].

### 2.4. A Distinct Set of UL18 Variants Associate with Metrics of the Burden of HCMV in Australian RTR

*UL18* variants which align with levels of antibodies reactive with HCMV lysate or IE-1 in RTR are summarized in Table 3. At position 32, 8/14 RTR carried only D, 5/14 carried both D and G and 1/14 only carried G. One RTR carrying HCMV encoding D, H and N at position 32 was excluded from analyses. RTR with the G variant had higher levels of antibodies reactive with HCMV lysate (*p* = 0.02) and IE-1 (*p* = 0.01).

Variants at positions 248, 250 and 252 were always carried together. Accordingly, individuals carrying N at position 248, A at position 250 or D at position 252 had elevated HCMV IE-1 antibody levels (Table 3, *p* = 0.002, *p* = 0.02 and *p* = 0.006, respectively). HCMV lysate antibodies followed a similar trend but no differences were statistically significant.

A distinct subset of *UL18* variants associated with altered T-cell (IFNγ) responses to HCMV and/or proportions of γδ T-cells with the Vδ2^−^ phenotype, previously associated with a high burden of HCMV [34]. Associations are summarized in Table 4. At position 29, 8/15 RTR carried only G and 7/15 carried both G and H. RTR carrying H had higher responses to HCMV lysate (*p* = 0.048).

Positions 191 and 196 demonstrated partial linkage disequilibrium. Accordingly, RTR with only K at position 191 or only Q at position 196 had elevated numbers of HCMV pp65-specific T-cells (Table 4, *p* = 0.02 and *p* = 0.004). Carriage of K at position 191 and Q at position 196 was also associated with expanded populations of Vδ2^−^ γδ T-cells (Table 4, *p* = 0.0082 and *p* = 0.002, respectively).

At position 265, 2/15 RTR carried F, 7/15 RTR carried F and L and 6/15 RTR carried only L. RTR with only L at position 265 had elevated numbers of HCMV pp65-specific T-cells (Table 4, *p* = 0.03).

### 2.5. Amino Acid Haplotypes

As several variants were co-expressed in diverse samples, haplotype analyses were performed using fastPhase. We included all 15 UL18 loci that were associated with any metrics of the burden of HCMV as described above. Table 5 shows 24 putative haplotypes with frequencies of ≥1%, designated UL18-1 to UL18-24 in descending order of frequency. The 24 haplotypes account for 68% of all samples. Frequencies of haplotypes did not differ between Australia and Indonesia, buffy coat and saliva samples or adults and congenital samples (data not shown). However, UL18-1 (present in 15% of all samples) encoded the variants L12M, M191K, E196Q, F265L and S318N which aligned with altered HCMV metrics in PWH or RTR.

The haplotypes confirm the co-carriage of variants in close proximity (191Kwith 196Q, 29H with 31Y, 181E with 185T). 248N, 250A and 252D always occurred together (see Table 5). S318N and T107A (See Figure 2 and Figure 3) were also carried together, with N at 318 and A at 107 in haplotypes UL18-6, -16, and -23 associated with low antibody levels in PWH and S at 318 and T at 107 in haplotypes UL18-9 and-10 associated with high antibody levels.

## 3. Discussion

In this study, we identify nonsynonymous variants in the *UL18* gene by sequencing HCMV directly from clinical samples. Compared to our analyses of *US28* and *UL111a* in the same samples [29,35], *UL18* had a higher amount of variation. Several variations were donor group specific, but eight were present in all cohorts and all sample types. This included four alleles that affected HCMV metrics, located within the α1, α2, α3 and the stalk domains of UL18. Variants D248N, V250A and E252D (N/A/D) are located in the α3 domain and appeared together in haplotypes generated by fastPhase (Table 5). Chen et al., 2016 [36] linked these variants with effective control of viral dissemination by LIR-1^+^ NK cells, suggesting altered binding kinetics with LIR-1. Here, we demonstrated that the N/A/D variant haplotype is common in clinical isolates of HCMV. However, the N/A/D variant haplotype was associated with elevated antibody levels in Australian RTR who were recruited > 2 years after transplantation. The simplest explanation is that one or more of these sites marks a *UL18* variant beneficial to the virus, driving higher antibody responses to compensate for poor NK cell function. The paradox may arise if NK cells are not the key to the control of HCMV in RTR. NK cells may be more important in PWH, where the N/A/D variant haplotype existed but did not significantly alter humoral responses to HCMV.

In RTRs, variants encoding the minor allele at G29H, M191K, E196Q or F265L were associated with higher T-cell responses to HCMV lysate and pp65 or larger populations of the HCMV induced Vδ2^−^ γδ T-cells, both markers of a higher burden of HCMV. However, we note no differences in their HCMV DNA levels assessed at the time of sequencing (*p* > 0.3, data not shown).

Variants encoding the minor allele at S318N or A107T were common, present in all groups, and the alleles were carried together in haplotypes UL18-9 and -10. Amongst PWH, N at 318 and A at 107 associated with lower levels of HCMV-reactive antibodies over the first few months on ART. Levels of HCMV-reactive antibodies increase on ART, stabilizing between 6 and 12 months [37]. This rise was only seen in PWH with S at 318 and T at 107. As neither variant was associated with differences in CD4^+^ T-cell counts, they may not affect immune recovery. However, they may affect the burden of HCMV such that it rises on ART in PWH with HCMV encoding S at 318 and T at 107, explaining their higher humoral responses to HCMV (lysate and IE-1) and higher CRP and soluble IFNαR levels (Figure 2 and Figure 3). It is also possible that one of these variants affects the immunoregulatory properties of UL18. Position 318 lies in the stalk domain and could affect cell surface expression of UL18 [38]. Unfortunately, we were unable to compare the HCMV DNA levels as sequences were obtained after 0–3 months on ART. It is remarkable that several variations were associated with lower antibody responses in the Indonesian PWH, but not in Australian RTR. This suggests that UL18 affects responses that are not compromised by the immunosuppressive effects of HIV. CD4^+^ T-cell counts were low at V0 [55 (2–199) cells/μl] [39], so modulation of NK cell function may be critical. JakCCANDO patients with abundant NKG2C^+^ and LIR-1^+^ NK cells displayed lower levels of HCMV reactive antibody [40], so it will be important to address bi-directional interactions between the expression of these receptors and replicating HCMV. UL18 may induce IFNγ and IFNα production by NK cells and T-cells [26], but the role of LIR-1 remains unclear as there is evidence of UL18 stimulating LIR-1^-^ NK cells [41].

In conclusion; we have shown that *UL18* is highly variable, with many nonsynonymous variants existing in patients with high burdens of HCMV. Distinct variants align with altered levels of HCMV-reactive antibody in RTR and PWH, and with T-cell responses assessed in RTR. These data highlight the need to dissect the role of the persistent burden of HCMV as this interacts bidirectionally with the immune responses measured. Studies addressing the effects of the variations identified here on LIR-1 binding and NK cell function would have both clinical and scientific value.

## 4. Materials and Methods

### 4.1. RTR and Healthy Controls from Perth, Western Australia

Eighty-two RTR were recruited from renal clinics at Royal Perth Hospital (Western Australia). Inclusion criteria were clinical stability greater than two years after transplant, no HCMV disease or reactivation within six months of sample collection and no current anti-viral treatment. RTR infected with hepatitis B or C were excluded. Ethics approval was obtained from Royal Perth Hospital Human Research Ethics Committee (approval number: EC 2012/155) and endorsed by Curtin University Human Research Ethics Committee (approval numbers: HRE16-2015 and HRE2021-0044). Participants provided written informed consent [31]. 

### 4.2. HIV Patients from Jakarta, Indonesia

The JakCCANDO Project is a comprehensive survey of clinical and immunological responses to ART undertaken in Cipto Mangunkusumo Hospital’s outpatient clinic (Jakarta, Indonesia) [33]. Eighty-two ART-naïve HIV patients were enrolled during 2013–2014 with <200 CD4^+^ T-cells/µl. The study was approved by Universitas Indonesia, Cipto Mangunkusumo Hospital and Curtin University ethics committees. Written consent was obtained from each subject. Samples were collected before ART initiation (V0) and at months 1, 3, 6 and 12 (V1, V3, V6, V12). Plasma HIV RNA loads were determined using AmpliPrep/COBAS^®^ TaqMan^®^ HIV-1 Tests (version 2.0) and CD4^+^ T-cell counts were determined using standard flow cytometric techniques [32].

### 4.3. Australia Neonates

Four de-identified samples of urine from congenitally infected babies were provided by the Department of Microbiology, PathWest Laboratory Medicine WA. Samples were collected between 1-13 days of life and all four had detectable HCMV DNA when assessed by routine hospital assays. Two neonates had symptomatic infections. One had hepatitis attributed to HCMV that spontaneously resolved without antiviral therapy. Another had bilateral sensorineural hearing loss, other central nervous system and lymphatic abnormalities and required antiviral therapy. 

### 4.4. Extraction and Detection of HCMV DNA

DNA was extracted from saliva, buffy coat or urine using FavorPrep Blood Genomic DNA Extraction Mini Kits (Favorgen, Ping-Tung, Taiwan) and stored at 80 °C. HCMV was detected using an in-house qPCR assay with primers targeting the *UL54* gene (HCMV DNA polymerase) [42].

### 4.5. Targeted Whole Gene Amplification

Primers targeting *UL18* were designed using Geneious 8.1.9 (https://www.geneious.com (accessed on 27 December 2018)) (*UL18*: 5′-3′: F- GAAGATAGGAGGGGTCAAAACGCGG, R- GAAGATAGGAGGGGTCAAAACGCGG). Reactions were performed in a total volume of 20 µl containing 0.4 µl of MyTaq HS DNA polymerase (Bioline, Meridian Bioscience, Cincinnati, OH, USA), 4 µl of MyTaq reaction buffer, 0.8 µl of 10uM primers (Sigma-Aldrich, Australia) and 5 µl of DNA diluted 1:2. Cycling conditions were 1 min at 95 °C followed by 30 cycles of 15 s at 95 °C, 15 s 60 °C and 1.5 min at 72 °C followed by a final extension step of 7 min at 72 °C. Amplicons were purified prior to preparation of DNA libraries using MO BIO Laboratories Inc UltraClean PCR Clean-Up Kit (Qiagen, Hilden, Germany).

### 4.6. Preparation of Ion Ampliseq™ DNA Libraries

Libraries were prepared using an Ion Ampliseq™ Library Kit 2.0 with halved reaction volumes and a total of 10 ng of template nucleic acid. The targets were amplified for 30 cycles with an anneal/extension time of 4 minutes per cycle. During library purification, ethanol was freshly prepared at 75% concentration. Libraries were quantified using a High-Sensitivity DNA Kit on a Bioanalyzer 2100 (Agilent, Santa Clara, CA, USA).

### 4.7. Libraries Were Sequenced Using an Ion Proton Sequencer

Barcoded sample libraries were diluted in low Tris-EDTA (Thermo Fisher Scientific, Waltham MA, USA) to reach a final concentration of 100 pmol/L, and equal volumes of each were pooled. The pooled libraries then underwent template preparation on an Ion Chef System and were loaded onto Ion P1 v3 sequencing chips using an Ion PI Hi-Q Chef Kit (Thermo Fisher Scientific). Semiconductor sequencing was performed on an Ion Proton Sequencer using an Ion PI Hi-Q Sequencing Kit (Thermo Fisher Scientific) [43].

### 4.8. Immunological Assessments of HCMV

Plasma stored in 80 °C were assessed for HCMV-reactive IgG titers using in-house ELISAs based on a lysate of fibroblasts infected with HCMV AD169 or IE-1 protein (Miltenyi Biotech, Cologne, Germany). Results are presented as arbitrary units (AU) based on a standard plasma pool, allowing comparisons between people but not between antigens.

Peripheral blood mononuclear cells (PBMC) isolated by Ficoll density centrifugation were stored in liquid nitrogen. PBMC were used to assess T-cell responses to HCMV lysate and a peptide pool derived from pp65 (JPT Peptide Technologies; Berlin, Germany) via ELISpot assay. These antigens and peptide pools are known to stimulate CD4 and CD8 T-cell responses. PBMC was also used to assess populations of Vδ2- γδ T-cells by flow cytometry as these are elevated in HCMV-seropositive RTR [34].

### 4.9. Data Analysis

Sequences were mapped to the Toledo reference (GenBank: GU937742.1) using the tmap tool within the Torrent Suite v 5.10. BAM files mapped to Toledo were loaded into proprietary software, Visual Genomics Analysis Suite (VGAS) (http://www.iiid.com.au/software/vgas. Variants were called if they occurred at a frequency of greater than 10% and had a minimum of 50 reads. VGAS was also utilized to identify changes in protein sequence.

Amino acid haplotypes and their estimated frequencies were determined using the default parameters of the fastPHASE algorithm with the exception that haplotypes were sampled an additional 5000 times [44]. Haplotypes with a population frequency less than 1% were excluded from analyses. Haplotypes are labelled UL18-1 to UL18-29 in descending order of their frequencies.

### 4.10. Statistical Analyses

Continuous data was analyzed with Mann–Whitney non-parametric statistics and categorical data was analyzed with Chi-squared or Fisher’s exact tests, as appropriate, using GraphPad Prism version 8 for Windows (Graphpad Software, La Jolla, CA, USA).

## Figures and Tables

**Figure 1 ijms-23-12911-f001:**
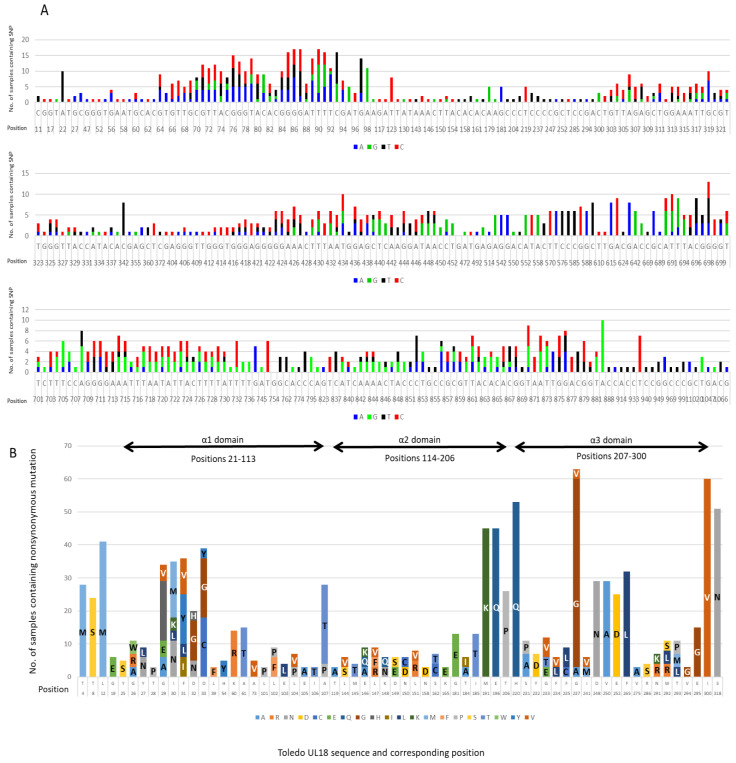
(**A**) **All nucleotide variations identified in HCMV *UL18* sequenced in 60 samples.** Variations are displayed in reference to HCMV Toledo. Blue bars represent A, green bars represent G, black bars represent T, and red bars represent C. (**B**) **All nonsynonymous mutations identified in HCMV *UL18* sequenced in 60 samples.** Variations are displayed in reference to HCMV Toledo. Amino acids are represented by their one-letter codes. Each variation presented was found in at least 3 samples. The height of the bars represents the number of samples carrying the variation.

**Figure 2 ijms-23-12911-f002:**
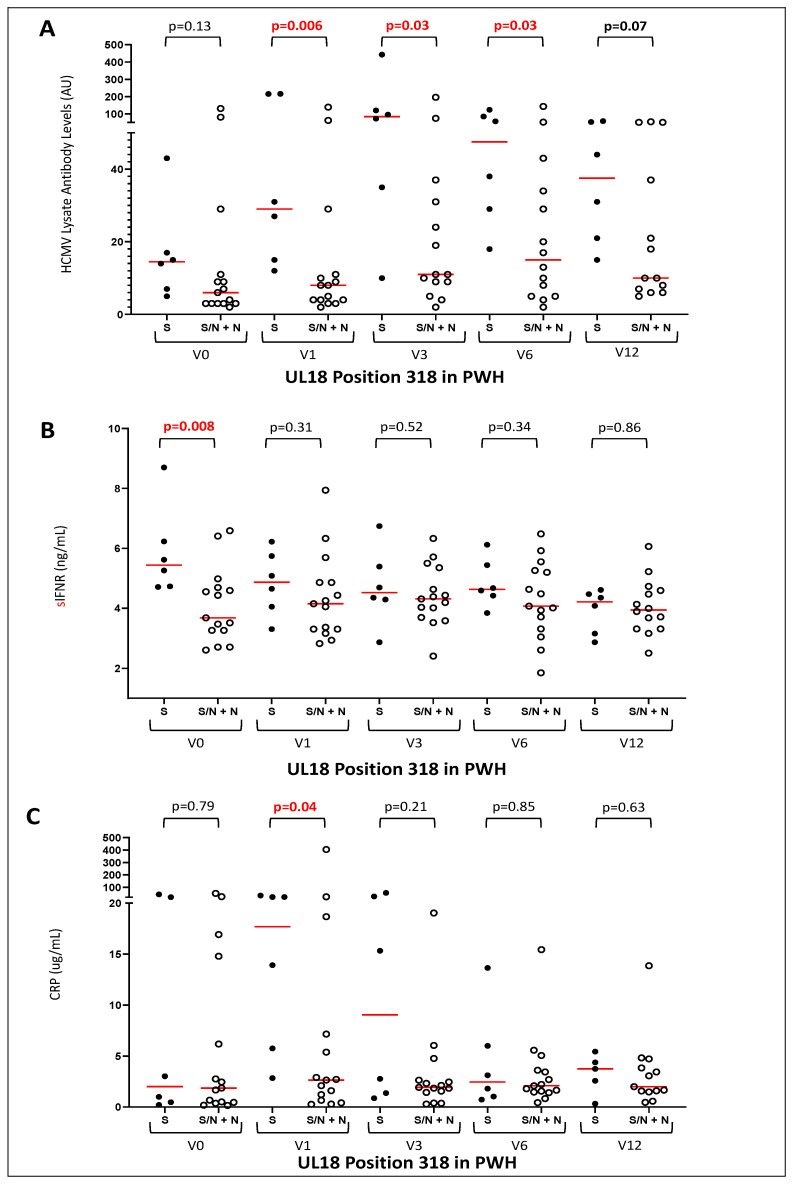
**PWH with S318N carried lower levels of HCMV-reactive antibodies and/or inflammatory biomarkers over 6 months on ART.** (**A**) comparison of HCMV lysate-reactive antibodies in PWH with S at position 318 and those with either S/N or N over time on ART. (**B**) plasma soluble IFNαR levels in PWH with HCMV with S at position 318 versus those with S/N or N; (**C**) plasma CRP levels in PWH with HCMV with S at position 318 versus those with S/N or N. V0 = before ART; V1, V3, V6, V12 = 1, 3, 6 or 12 months on ART resp. Red lines mark median values.

**Figure 3 ijms-23-12911-f003:**
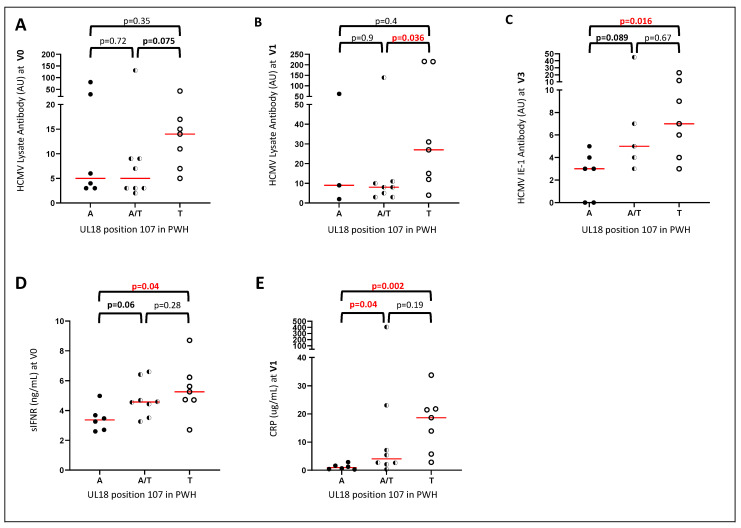
**PWH carrying A107T had higher levels of HCMV reactive antibody levels and inflammatory biomarkers over time.** (**A**) HCMV lysate-reactive antibodies at V0 (0 months on ART) with A, A/T or T at position 107; (**B**) HCMV lysate-reactive antibodies at V1 (1 month on ART) with A, A/T or T at position 107; (**C**) HCMV IE-1-reactive antibodies at V3 (3 months on ART) with A, A/T or T at position 107; (**D**) plasma soluble IFNαR levels at V0 with A, A/T or T at position 107; (**E**) plasma CRP levels at V1 with A, A/T or T at position 107. Red lines mark median values.

**Table 1 ijms-23-12911-t001:** UL18 protein variants distinct from Toledo were found in all groups.

Residue Position	Toledo Ref	Neonates *n* = 4	Adults *n* = 55	Australian *n* = 28	Indonesian *n* = 27	Buffy Coat *n* = 31	Saliva *n* = 24
**4**	**T**	TM	TM	TM	TM	TM	TM
**8**	**T**	TS	TS	TS	TS	TS	TS
**12**	**L**	LM	LM	LM	LM	LM	LM
**19**	**G**	G	G**E**	G**E**	G	GE	GE
**25**	**Y**	Y	Y**S**	Y**S**	Y	YS	YS
**26**	**G**	G	G**ARW**	G**A**RW	GRW	GAW	GARW
**27**	**Y**	Y	Y**NL**	YN**L**	YN	YL	YNL
**28**	**T**	T	T**P**	T**P**	T	TP	TP
**29**	**G**	G	G**AEHV**	GAEH**V**	GAEH	GAEHV	GAEHV
**30**	**I**	I	I**NLKMT**	IN**LK**M**T**	INM	INLKM	INLKMT
**31**	**F**	F	F**ILYV**	FILYV	FILYV	FILYV	FILYV
**32**	**D**	D	D**NGH**	DNGH	DNGH	DNGH	DNGH
**33**	**D**	D	D**CGY**	DCGY	DCGY	DCGY	DCGY
**39**	**L**	L	L**F**	LF	LF	LF	LF
**54**	**H**	H	H**Y**	HY	HY	HY	HY
**60**	**K**	KR	KR	KR	KR	KR	KR
**61**	**A**	AT	AT	A**T**	A	AT	AT
**73**	**A**	A	A**V**	AV	AV	AV	AV
**101**	**L**	L	L**P**	L**P**	L	L	L**P**
**102**	**L**	L	L**FP**	LF**P**	LF	LFP	LFP
**103**	**E**	EL	EL	E**L**	E	EL	EL
**104**	**L**	L	L**V**	LV**P**	LV	LP	LPV
**105**	**E**	E	E**A**	E**A**	E	EA	EA
**106**	**I**	I	I**T**	I**T**	I	IT	IT
**107**	**A**	A	A**PT**	A**P**T	AT	APT	APT
**119**	**T**	T	T**A**	T	T**A**	TA	TA
**144**	**L**	L	L**SV**	L**SV**	L	LSV	LSV
**145**	**M**	M	M**T**	M**T**	M	MT	MT
**146**	**E**	E	E**AQK**	E**AQK**	E	EAQK	EAQK
**147**	**L**	L	L**RVF**	L**RVF**	L	LRV	LRV**F**
**148**	**K**	K	K**NQ**	K**NQ**	K	KNQ	KNQ
**149**	**D**	D	D**ES**	D**ES**	D	DES	DES
**150**	**N**	N	N**DC**	N**DC**	N	ND	NC
**151**	**L**	L	L**RV**	LR**V**	LR	LRV	LRV
**158**	**N**	N	N**D**	ND	ND	ND	ND
**162**	**S**	S	S**CT**	SCT	SCT	ST	SCT
**165**	**K**	K	K**ET**	KE	KE	K	KE
**181**	**G**	G	G**E**	GE	GE	GE	GE
**184**	**T**	TAI	TAI	T**A**	T**I**	TA**I**	TA
**185**	**I**	I	I**T**	IT	IT	IT	IT
**191**	**M**	MK	MK	MK	MK	MK	MK
**196**	**E**	EQ	EQ	EQ	EQ	EQ	EQ
**206**	**T**	T	T**P**	TP	TP	TP	TP
**220**	**H**	H	H**QT**	HQ**T**	HQ	HQ	HQ**T**
**231**	**S**	S	S**APT**	SAP	SAP	SAP	SAP
**232**	**Y**	Y	Y**DS**	Y**DS**	Y	YDS	YDS
**233**	**G**	G	G**ETV**	G**E**TV	GTV	GETV	GETV
**234**	**F**	F	F**LV**	F**LV**	F	FLV	FLV
**235**	**F**	F	F**CL**	F**C**L	FL	FCL	FCL
**237**	**G**	G	G**AV**	G**AV**	G	GAV	GAV
**241**	**I**	I	I**MTV**	I**MTV**	I	IMV	IM**T**V
**248**	**D**	D	D**N**	DN	DN	DN	DN
**250**	**V**	V	V**A**	VA	VA	VA	VA
**252**	**E**	E	E**D**	ED	ED**Q**	EDQ	E
**265**	**F**	F	F**L**	FL	FL	FL	FL
**275**	**V**	V	V**A**	VA	VA	VA	VA
**286**	**R**	R	R**S**	R**S**	R	R**S**	R**S**
**291**	**N**	N	N**RKT**	N**RKT**	N	NRK	NRK**T**
**292**	**W**	W	W**RLS**	W**R**L**S**	WL	WRLS	WRLS
**293**	**T**	T	T**LMP**	T**LM**P	TP	TLMP	TLMP
**294**	**V**	V	V**G**	V**G**	V	VG	VG
**295**	**E**	EG	EG	EG	EG	EG	EG
**300**	**I**	I	I**V**	IV	IV	IV	IV
**318**	**S**	SN	SN	SN	SN	SN	SN

Amino acids encoded by non-synonymous mutations are displayed in reference to Toledo. Changes unique to a group are in bold. All mutations reported were present in at least 3 samples.

**Table 2 ijms-23-12911-t002:** *UL18* variants are associated with altered HCMV-reactive antibody levels in PWH.

L12M									
	L (*n* = 3)			L/M + M (*n* = 18)			*p*-Value ^a^		
V12, HCMV lysate antibody (AU)	537 (213–578)			169 (49–530)			**0.048**		
V12, HCMV IE-1 antibody (AU)	12 (6–36)			5 (0–73)			0.21		
V6, CD4^+^ T-cells /ul	329 (324–394)			188 (38–463)			**0.024**		
V12, CD4^+^ T-cells/ul	377 (335–763)			263 (66–655)			**0.064**		
**F31Y**									
	**F (*n* = 17)**			**F/Y (*n* = 4)**					
V1, HCMV lysate antibody (AU)	12 (4–216)			3 (2–10)			**0.012**		
V1, HCMV IE-1 antibody (AU)	4 (0–26)			3 (3–10)			0.81		
**H54Y**									
	**H (*n* = 18)**			**H/Y +Y (*n* = 3)**					
V3, HCMV lysate antibody (AU)	31 (2–443)			10 (9–11)			0.18		
V3, HCMV IE-1 antibody (AU)	5.5 (0–45)			3 (0–3)			**0.028**		
**G181E (in LD with I185T)**									
	**G (*n* = 11)**		**G/E (*n* = 7)**		**E (*n* = 3)**				
V12, HCMV lysate antibody (AU)	14 (5–53)		29 (7–54)		21 (15–58)		0.43 ^b^	0.90 ^c^	0.37 ^d^
V12, HCMV IE-1 antibody (AU)	4.5 (1–19)		16.5 (6–73)		3 (0–12)		**0.038**	0.14	0.54
**M191K**									
	**M (*n* = 4)**		**M/K (*n* = 6)**		**K (*n* = 10)**				
V3, HCMV lysate antibody (AU)	65 (24–443)		11 (5–31)		11 (2–196)		**0.03** ** ^e^ **	0.72 ^f^	0.22 ^g^
V3, HCMV IE-1 antibody (AU)	9 (5–23)		5 (3–12)		4 (0–45)		0.34	0.73	0.11
**S231A**									
	**S (*n* = 18)**			**S/A (*n* = 3)**					
V12, HCMV lysate antibody (AU)	17 (5–58)			51 (44–54)			**0.05**		
V12, HCMV IE-1 antibody (AU)	5 (0–73)			12 (10–36)			0.11		

Comparison of HCMV lysate or IE-1 antibody levels with variants of *UL18* in PWH. ^a^ Mann–Whitney T-test. ^b^ G vs. G/E, ^c^ G/E vs. E, ^d^ G vs. E. ^e^ M vs. M/K, ^f^ M/K vs. K, ^g^ M vs. K. L/M indicates the carriage of both L and M.

**Table 3 ijms-23-12911-t003:** *UL18* variants are associated with altered HCMV-reactive antibody levels in RTR.

D32G			
	D (*n* = 7)	D/G + G (*n* = 5)	*p*-Value ^a^
HCMV lysate antibody (AU)	412 (1–1292)	3241 (554–5582)	**0.02**
HCMV IE-1 antibody (AU)	51 (5–178)	446 (103–1463)	**0.01**
**D248N**			
	**D (*n* = 7)**	**D/N + N (*n* = 6)**	
HCMV lysate antibody (AU)	411 (1–3241)	1446 (117–5582)	0.18
HCMV IE-1 antibody (AU)	51 (5–103)	312 (92–1463)	**0.002**
**V250A**			
	**V (*n* = 6)**	**V/A + A (*n* = 7)**	
HCMV lysate antibody (AU)	570 (1–3241)	1292 (117–5582)	0.37
HCMV IE-1 antibody (AU)	59 (5–103)	178 (32–1463)	**0.02**
**E252D**			
	**E (*n* = 8)**	**E/D + D (*n* = 5)**	
HCMV lysate antibody (AU)	483 (1–3241)	1599 (117–5582)	0.13
HCMV IE-1 antibody (AU)	59 (5–143)	446 (92–1463)	**0.006**

Comparison of HCMV lysate or IE-1 antibody levels with variants of *UL18* in RTR. ^a^ Mann–Whitney T-test.

**Table 4 ijms-23-12911-t004:** *UL18* variants are associated with altered HCMV-induced T-cells in RTR.

G29H			
	G (*n* = 6)	G/H (*n* = 6)	*p*-value ^a^
HCMV lysate-reactive T-cells	20.5 (0–78)	205.5 (7–938)	**0.048**
pp65-reactive T-cells	474 (0–1989)	316 (0–1235)	0.73
Vδ2^−^ γδ T-cells	7.7 (0.2–14.3)	3.7 (0.3–14.3)	0.66
**M191K**			
	**M + M/K (*n* = 6)**	**K (*n* = 7)**	
HCMV lysate-reactive T-cells	7 (0–938)	63 (0–509)	0.78
pp65-reactive T-cells	5 (0–524)	759 (52–1989)	**0.02**
Vδ2^−^ γδ T-cells	2.04 (0.19–7.70)	11.20 (3.31–14.30)	**0.0082**
**E196Q**			
	**E + E/Q (*n* = 7)**	**Q (*n* = 6)**	
HCMV lysate-reactive T-cells	23.5 (0–938)	70.5 (0–509)	0.67
pp65-reactive T-cells	28.5 (0–524)	997 (283–1989)	**0.004**
Vδ2^−^ γδ T-cells	2.21 (0.19–7.70)	11.75 (4.2–14.3)	**0.002**
**F265L**			
	**F + F/L (*n* = 7)**	**L (*n* = 5)**	
HCMV lysate- reactive T-cells	40 (0–938)	63 (0–509)	0.97
pp65-reactive T-cells	52 (0–1235)	759 (283–1989)	**0.03**
Vδ2^−^ γδ T-cells	3.3 (0.2–14.3)	9.8 (1.9–14.3)	0.14

Comparison of HCMV-specific T-cells or populations of Vδ2^−^ γδ T-cells with variants of *UL18* in RTR. ^a^ Mann–Whitney T-tests.

**Table 5 ijms-23-12911-t005:** Several *UL18* variants are frequently carried together.

**Position**	12	29	31	32	54	107	181	185	191	196	231	248	250	252	265	318	Frequency
**Toledo**	L	G	F	D	H	A	G	I	M	E	S	D	V	E	F	S	
**Variant**	M	H	Y	G	Y	T	E	T	K	Q	A	N	A	D	L	N	
**Haplotypes**																	
UL18-1	M	G	F	D	H	A	G	I	K	Q	S	D	V	E	L	N	0.15
UL18-2	L	G	F	D	H	A	G	I	K	Q	S	N	A	D	L	N	0.05
UL18-3	L	G	F	D	H	A	G	I	K	Q	S	D	V	E	L	N	0.04
UL18-4	M	G	F	D	H	A	G	I	M	E	S	D	V	E	L	N	0.03
UL18-5	M	G	F	D	H	A	G	I	K	Q	S	N	A	D	L	N	0.03
UL18-6	M	G	F	D	H	T	G	I	M	E	S	D	V	E	L	N	0.03
UL18-7	M	G	F	D	H	T	G	I	M	E	S	D	V	E	L	S	0.03
UL18-8	L	G	F	G	H	A	G	I	M	E	S	N	A	D	L	N	0.03
UL18-9	L	G	F	D	H	T	G	I	M	E	S	D	V	E	L	S	0.03
UL18-10	L	G	F	D	H	T	E	T	K	Q	S	D	V	E	L	S	0.03
UL18-11	L	H	Y	G	H	A	G	I	K	Q	S	D	V	E	L	N	0.03
UL18-12	L	G	F	D	H	A	G	I	M	E	S	D	V	E	L	N	0.03
UL18-13	L	G	F	D	H	A	G	I	M	E	S	N	A	D	L	N	0.03
UL18-14	L	G	F	D	H	T	G	I	M	E	S	N	A	D	L	S	0.02
UL18-15	M	G	F	D	H	A	G	I	M	E	S	N	A	D	L	N	0.02
UL18-16	M	G	F	D	H	T	G	I	K	Q	S	N	A	D	L	N	0.02
UL18-17	M	G	F	D	H	T	E	T	K	Q	S	N	A	D	L	S	0.02
UL18-18	L	H	Y	D	H	A	G	I	K	Q	S	D	V	E	L	N	0.02
UL18-19	M	G	F	D	H	A	G	I	K	Q	S	N	A	D	L	N	0.02
UL18-20	M	G	F	D	Y	A	E	T	K	Q	S	D	V	E	L	N	0.02
UL18-21	M	G	F	G	H	A	G	I	K	Q	S	N	A	D	L	N	0.01
UL18-22	L	G	F	D	H	A	G	I	K	Q	S	N	A	D	L	N	0.01
UL18-23	L	G	F	D	H	T	G	I	M	E	S	D	V	E	L	N	0.01
UL18-24	L	H	Y	D	H	A	G	I	K	Q	S	N	A	D	L	N	0.01

Grey shading represents residues differing from the Toledo reference.

## Data Availability

Amplicon sequence data have been deposited in NCBI under accession no. SAMN21506830 to SAMN21506889.

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
