# Peer review of "Variants of HCMV UL18 Sequenced Directly from Clinical Specimens Associate with Antibody and T-Cell Responses to HCMV"

_ijms, 2022, doi:10.3390/ijms232112911_

Round 1

Reviewer 1 Report

The manuscript titled “Variants of HCMV UL18 sequenced directly from clinical specimens associate with antibody and T-cell responses to HCMV” by Waters et al. investigated nonsynonymous variants in the HCMV UL18 gene from clinical samples.

I believe this study provides the necessary information in this field. However, there are concerns regarding this study. Through previous studies, the authors analyzed variants of UL28 and UL111a and reported the correlation between immune responses and the variants (Waters et al. 2021&2022). When UL18 variants and UL28 or UL111a variants are simultaneously generated in the same patient sample, there is a lack of comprehensive analysis and consideration of the results. Similar analyzes were performed with different three genes, but now it is necessary to analyze the previous results and this new result by combining them. Although each gene may have unique functions and sites of action, it is questionable whether it is reasonable to conclude without a comprehensive analysis of mixed variants. In addition, studies of the immunological mechanisms at the molecular level should be carried out to see how specific genetic mutations affect immunological changes. Overall, despite the importance of the research, new insights are lacking.

This is a personal opinion that may not be important, but the content of this manuscript's introduction is similar to that of their previous paper (Waters et al. 2022. IJMS). I think it is undesirable to write articles that are similar to recently published papers in the same journal.

Author Response

see attached document

Reviewer 2 Report

A key previous publication documenting variability of CMV (genome wide) is not cited. Aside from that missing context, this paper provides careful documentation of variability of UL18 and is a worthwhile contribution to the literature.  

Renzette N, Pokalyuk C, Gibson L, Bhattacharjee B, Schleiss MR, Hamprecht K, Yamamoto AY, Mussi-Pinhata MM, Britt WJ, Jensen JD, Kowalik TF. Limits and patterns of cytomegalovirus genomic diversity in humans. Proc Natl Acad Sci USA. 112(30):E4120-8 (2015).

Author Response

Please note - this also replies to reviewer 1

Round 2

Reviewer 1 Report

I have no idea what it would mean to infer the biological function of individual genetic variations in the absence of reverse genetics research. How can the authors conclude that the observed results were caused by genetic mutations? After computer analysis, it must be demonstrated experimentally. By making a specific genetic mutation in the virus gene, the immunological role of the virus must be shown. If not, I believe this manuscript should be submitted to a bioinformatics-specific journal. It is unlikely that I will be able to thoroughly evaluate this manuscript with my current abilities. I apologize, but I hope editor would find an additional reviewer.